# RECURRENT RELATIONAL NETWORKS FOR COMPLEX RELATIONAL REASONING

## ABSTRACT

Humans possess an ability to abstractly reason about objects and their interactions, an ability not shared with state-of-the-art deep learning models. Relational networks, introduced by Santoro et al. (2017), add the capacity for relational reasoning to deep neural networks, but are limited in the complexity of the reasoning tasks they can address. We introduce *recurrent relational networks* which increase the suite of solvable tasks to those that require an order of magnitude more steps of relational reasoning. We use recurrent relational networks to solve Sudoku puzzles and achieve state-of-the-art results by solving 96.6% of the hardest Sudoku puzzles, where relational networks fail to solve any. We also apply our model to the BaBi textual QA dataset solving 19/20 tasks which is competitive with state-of-the-art sparse differentiable neural computers. The recurrent relational network is a general purpose module that can augment any neural network model with the capacity to do many-step relational reasoning.

## 1 INTRODUCTION

A central component of human intelligence is the ability to abstractly reason about objects and their interactions (Spelke et al., 1995; Spelke & Kinzler, 2007). An abstract and relational framework is required for problem-solving, often requiring a strictly methodical approach. This paper introduces a composite function, the *recurrent relational network*, to serve as a modular component for higher-level relational reasoning in artificially intelligent agents or systems.

The popular puzzle game Sudoku serves as an illustrative example of methodical and relational problem solving. In it, 81 cells are arranged in a 9x9 grid, which must be filled with digits 1 to 9 so each digit appears exactly once in each row, column and 3x3 non-overlapping box. Sudokus are harder when less initial clues, or filled cells, are given.[1] To solve it, we would reason about the puzzle in terms of its cells and inter-cell interactions over many steps, rather than the puzzle in its entirety. Following Santoro et al. (2017) we use the term "relational reasoning" for this object- and interaction-centric thinking. Note that relational reasoning is not strictly defined, and is not, for instance equal to relational or first order logic. Sudoku shares properties with other problems that also require complex, multi-step relational reasoning: evaluating game moves, logical deduction, predicting the future of physical systems, and automated planning and resource allocation, like creating timetables, scheduling taxis and designing seating plans.

State-of-the-art deep learning approaches fall short when faced with problems that require basic relational reasoning (Lake et al., 2016; Santoro et al., 2017). The relational network of Santoro et al. (2017) gave a first glimpse of how this kind of reasoning could be achieved. However, it is limited to performing a single relational operation, and was evaluated on datasets that require a maximum of three steps of reasoning. We focus on Sudoku as a main task, as it requires an order of magnitude more steps of relational reasoning than has previously been considered. We show in this paper that the recurrent relational network learns an iterative strategy that *generalizes*, that is, can solve any unseen Sudoku.

One key insight by Santoro et al. (2017) is to split the problem into two components: a perceptual front-end and a relational reasoning module. The task of the perceptual front-end is to recognize

---

[1]We invite the reader to solve the Sudoku in the appendix to appreciate the difficulty of solving a Sudoku in which 17 cells are initially filled.

objects in the raw input and output a representation of them. The task of the relational reasoning module is to reason about the objects and their interactions. Both modules are trained jointly end-to-end, with the relational reasoning module requiring the perceptual front-end to recognize and represent objects that it can reason about. In computer science parlance, the relational reasoning module implements an *interface*; it operates on a set of objects described with real valued vectors, and is differentiable. The abstraction that the interface provides allows us to consider and improve each side of it in isolation. For many domains very good perceptual front-ends have already been found. For example, for images or text, convolutional and recurrent neural networks respectively, are natural choices. In this paper we focus on and improve the relational reasoning side of that interface, as it is still in its infancy in state-of-the-art deep learning architectures. As long as the interface is complied with, the recurrent relational module that is developed in this paper will be compatible with any perceptual front-end. For a formal definition of the interface see the appendix.

Solving Sudokus computationally is *in itself* not a very laudable goal, as traditional symbolic and hand-crafted algorithms like constraint propagation and search can solve any Sudoku in fractions of a second. For a good explanation and code, see Norvig (2006). Many other symbolic algorithms exist that can also solve Sudokus, like dancing links (Knuth, 2000) and integer programming. These algorithms are superior in almost every respect but one: they don't comply with the interface, as they don't operate on a set of vectors, and they're not differentiable. As such they cannot be used in a combined model with a deep learning perceptual front-end.

There is a rich litterature on logic and reasoning in artificial intelligence and machine learning. Please see section 5 for a discussion of related work.

## 2    RECURRENT RELATIONAL NETWORKS

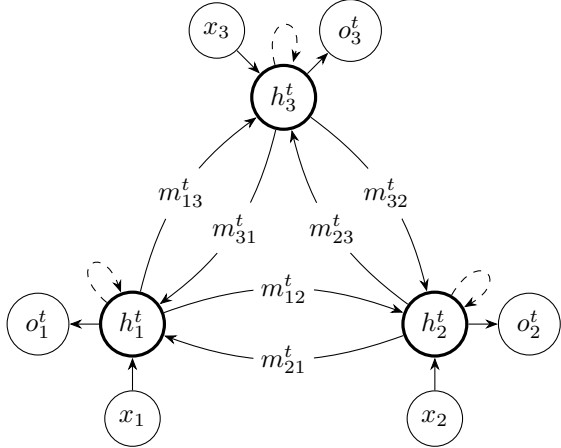

Figure 1: A *recurrent relational network* on a fully connected graph with 3 nodes. The nodes' hidden states $h_i^t$ are highlighted. The dashed lines indicate the recurrent connections. Subscripts denote node indices and superscripts denote steps $t$. For a figure of the same graph unrolled over 2 steps see the appendix.

Let's consider what a relational reasoning module might need in order to implement an elimination strategy to solve a Sudoku puzzle. Our intention is not to directly implement an elimination strategy, nor to restrict the network to it, but it will serve as a minimal requirement. The elimination strategy works by noting that if a certain cell is given as a "7", one can safely remove "7" as an option from other cells in the same row, column and box. If this is done for all cells, one might end up with cells that only have a single possible digit left. This digit could then be removed from the possible cell values in the same row, column and box, and so on. To implement this strategy, each cell needs to send a message to each other cell in the same row, column and box saying "I'm a 7, hence you can't also be a 7". Each cell should then consider all messages coming in, and update its own state. With the updated state each cell should send out new messages, and so forth.

We will formalize this by considering the Sudoku as a graph. The graph has $i \in \{1, 2, ..., 81\}$ nodes, one for each cell in the Sudoku. Each node has an edge to and from all nodes that is in the same row, column and box in the Sudoku. Each node has a feature vector $x_i$. As per the interface this set of feature vectors $\mathbf{x} = \{x_1, x_2, ..., x_{81}\}$ are the inputs to our relational reasoning module and would in general be the output of a perceptual front-end. For our Sudoku example each $x_i$ encodes the initial cell content (empty or given) and the row and column position. At each step $t$ each node has a hidden state vector $h_i^t$. We initialize this hidden state to the features, such that $h_i^0 = x_i$. At each step $t$, each node sends a message to each of its neighboring nodes. We define the message $m_{ij}^t$ from node $i$ to node $j$ at step $t$ by

$$m_{ij}^t = f\left(h_i^{t-1}, h_j^{t-1}\right) ,\tag{1}$$

where $f$, the message function, is a multi-layer perceptron (MLP). This allows the network to learn what kind of messages to send. Since a node needs to consider all the incoming messages we sum them with

$$m_{\cdot j}^t = \sum_{i \in N(j)} m_{ij}^t ,\tag{2}$$

where $N(j)$ are all the nodes that have an edge into node $j$, i.e. the nodes in the same row, column and box. Finally we update the node hidden state via

$$h_j^t = g\left(h_j^{t-1}, x_j, m_{\cdot j}^t\right) ,\tag{3}$$

where $g$, the node function, is another learned neural network. The dependence on the previous node hidden state $h_j^{t-1}$ allows the network to iteratively work towards a solution instead of starting with a blank slate at every step. Injecting the feature vector $x_j$ at each step like this allows the node function to focus on the messages from the other nodes instead of trying to remember the input.

The above equations for sending messages and updating node states define a recurrent relational network's core, and we now use it to train a neural network in a supervised manner to solve a Sudoku. At every step each node outputs a probability distribution over the digits 1-9 and we minimize the cross entropy between this output probability distribution and the target digit from the Sudoku solution. The output probability distribution $o_i^t$ for node $i$ at step $t$ is given by

$$o_i^t = \text{softmax}\left(r\left(h_i^t\right)\right) ,\tag{4}$$

where $r$ is a MLP that maps the node hidden state to the output logits. Given the target digit $y_i$ (1-9) for cell $i$, the cross-entropy node loss $l_i^t$ for node $i$ at step $t$ is

$$l_i^t = -\log o_i^t\left[y_i\right] ,\tag{5}$$

where the square brackets are used to indicate the $y_i$'th element of the vector. For a single Sudoku puzzle $\mathbf{x} = \{x_1, x_2, ..., x_{81}\}$ and its solution $\mathbf{y} = \{y_1, y_2, ..., y_{81}\}$ the total loss $\mathcal{L}(\mathbf{x}, \mathbf{y})$ is the sum of losses computed recurrently over all $I = 81$ nodes and $T$ steps,

$$\mathcal{L}(\mathbf{x}, \mathbf{y}) = \sum_{t=1}^{T} \sum_{i=1}^{I} l_i^t .\tag{6}$$

To train the network we minimize the total loss, with respect to the parameters of the functions $f$, $g$ and $r$ using stochastic gradient descent. See figure 1 for an example of the recurrent relational network on a fully connected graph with 3 nodes.

At test time we only consider the output probabilities at the last step, but having a loss at every step during training is beneficial. Since the target digits $y_i$ are constant over the steps, it encourages the network to learn a convergent algorithm. Secondly, it helps with the vanishing gradient problem. One potential issue with having a loss at every step is that it might force the network to learn a greedy algorithm that gets stuck in a local minima. However, the separate output function $r$ allows the node hidden states and messages to be different from the output probability distributions. As such, the network could use a small part of the hidden state for retaining a current best guess, which might remain constant over several steps, and other parts of the hidden state for running a non-greedy multi-step algorithm.

Sending messages for all nodes in parallel and summing all the incoming messages might seem like an unsophisticated approach that risk resulting in oscillatory behavior and drowning out the

important messages. However, since the receiving node hidden state is an input to the message function, the receiving node can in a sense determine which messages it wishes to receive. As such, the sum can be seen as an implicit attention mechanism over the incoming messages. Similarly the network can learn an optimal message passing schedule, by ignoring messages based on the history and current state of the receiving and sending node.

We have described our model from the example of solving Sudokus, but the model is in no way limited to Sudokus. In general, as per the interface, it takes as input a set of objects described by feature vectors and a set of edges detailing how the objects interacts. If the edges are unknown, the graph can be assumed to be fully connected. In this case the network will need to learn which objects interact with each other. If the edges have attributes, $e_{ij}$, the message function in equation 1 can be modified such that $m_{ij}^t = f\left(h_i^{t-1}, h_j^{t-1}, e_{ij}\right)$. If the output of interest is for the whole graph instead of for each node the output in equation 4 can be modified such that there's a single output $o^t = r\left(\sum_i h_i^t\right)$. The loss can be modified accordingly.

# 3 EXPERIMENTS

Code to reproduce the experiments can be found at *redacted for peer review*.

## 3.1 SUDOKU

We generate a dataset of 216,000 puzzles with an equal number of 17 to 34 givens from the collection of 49,151 unique 17-givens puzzles gathered by Royle (2014). We use the solver from Norvig (2006) to solve all the puzzles first. Then we split the puzzles into a test, validation and training pool. To generate the training, validation and test set, we sample puzzles from the respective pools, add between 0 to 17 givens from the solution, and swap the digits according to a random map per Sudoku, e.g. $1 \rightarrow 5, 2 \rightarrow 3$, etc.

We consider each of the 81 cells in the 9x9 Sudoku grid a node in a graph, with edges to and from each other cell in the same row, column and box. Denote the digit for cell $j$ $d_j$ (0-9, 0 if not given), and the row and column position $\text{row}_j$ (1-9) and $\text{column}_j$ (1-9) respectively. The node features are then $x_j = \text{MLP}\left(\left[\text{embed}\left(d_j\right); \text{embed}\left(\text{row}_j\right); \text{embed}\left(\text{column}_j\right)\right]\right)$ where each embed is a separate 16 dimensional learnable embedding and $[a; b]$ denotes the concatenation of $a$ and $b$. We don't use any edge features and we don't treat the cells with given digits in any special way. The message from $i$ to $j$ is $m_{ij}^t = \text{MLP}\left(\left[h_i^{t-1}; h_j^{t-1}\right]\right)$. The node hidden state is given by $h_j^t = \text{LSTM}\left(\text{MLP}\left(\left[x_j; m_{\cdot j}^t\right]\right)\right)$ where LSTM denotes a Long Short Term Memory cell (Hochreiter & Schmidhuber, 1997). The LSTM cell and hidden state is initialized to zero. The output function $r$ is a linear layer with ten outputs to produce the output logits $o_i^t$. All the MLP's are four layers with 96 nodes. The first 3 layers have ReLU activation functions and the last layer is linear. The LSTM also has 96 nodes. We run the network for 32 steps. We train the model for 300.000 gradient updates with a batch size of 252 using Adam with a learning rate of 2e-4 and L2 regularization of 1e-4 on all weight matrices.

Our network learns to solve 94.1% of even the hardest 17-givens Sudokus after 32 steps. For more givens the accuracy quickly approaches 100%. Since the network outputs a probability distribution for each step, we can visualize how the network arrives at the solution step by step. For an example of this see figure 2. In the first step, the network uses the elimination strategy to reduce the number of possible digits. For subsequent steps it assigns softer probabilities to the digits, and seems to try a number of different configurations. Once the solution is found it locks onto it and doesn't change.

To examine our hypothesis that multiple steps are required we plot the accuracy as a function of the number of steps. See figure 3. We can see that even simple Sudokus with 33 givens require upwards of 10 steps of relational reasoning, whereas the harder 17 givens continue to improve even after 32 steps. Figure 3 also shows that the model has learned a convergent algorithm. The model was trained for 32 steps, but seeing that the accuracy increased with more steps, we ran the model for 64 steps during testing. At 64 steps the accuracy for the 17 givens puzzles increases to 96.6%.

We compare our network to the relational network (Santoro et al., 2017). We train two relational networks: a node and a graph centric. The node centric corresponds exactly to a single step of our network. The graph centric approach is closer to the original relational network. It does one

Figure 2: Example of how the trained network solves part of a Sudoku. Only the first column of a full 9x9 Sudoku is shown for clarity. See appendix for the full Sudoku. Each cell displays the digits 1-9 with the font size scaled (non-linearly for legibility) to the probability the network assigns to each digit. We only show steps 0, 1, 4, 8 and 12 due to space constraints. Notice how the network eliminates the given digits 4 and 8 from the other cells in the first step. For this particular Sudoku the network converges to the solution after approximately 20 steps. Animations showing how the trained network solves Sodukos, including a failure case can be found at imgur.com/a/ALsfB.

step of relational reasoning as our network, then sums all the node hidden states. The sum is then

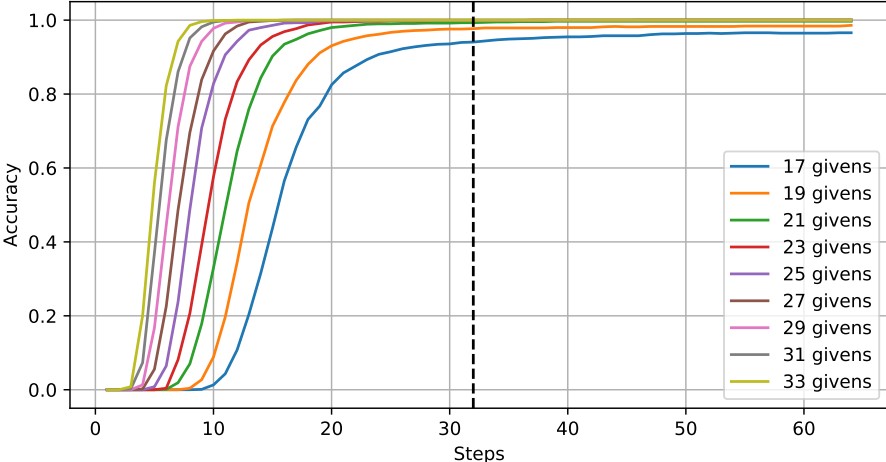

Figure 3: Accuracy of our trained network on Sudokus as a function of number of steps. Even simple Sudokus with 33 givens require about 10 steps of relational reasoning to be solved. The dashed vertical line indicates the 32 steps the network was trained for. The network appears to have learned a convergent relational reasoning algorithm such that more steps beyond 32 improve on the hardest Sudokus.

passed through a 4 layer MLP with $81 \cdot 9$ outputs, one for each cell and digit. The graph centric model has larger hidden states of 256 in all layers to compensate somewhat for the sum squashing the entire graph into a fixed size vector. Otherwise both networks are identical to our network. We could not get either of them to solve any Sudokus. Of the two, the node centric trained much faster and got considerably lower loss. The only difference between the node centric relational network and our model is the number of steps, yet the relational network fails to solve any Sudoku. This shows that multiple steps are crucial for complex relational reasoning. The graph centric has over 4 times as many parameters as our model (944,874 vs. 201,194) but performs even worse than the node centric. We also compare our network to other differentiable methods. See table 1. Our network outperforms loopy belief propagation, with parallel and random messages passing updates (Bauke, 2008). It also outperforms a version of loopy belief propagation modified specifically for solving Sudokus that uses 250 steps, sinkhorn balancing every two steps and iteratively picks the most probable digit (Khan et al., 2014). We also compare to learning the messages in parallel loopy BP as presented in Lin et al. (2015). We tried a few variants including a single step as presented and 32 steps with and without a loss on every step, but could not get it to solve any 17 given Sudokus. Finally we outperform Park (2016) which treats the Sudoku as a 9x9 image, uses 10 convolutional layers, iteratively picks the most probable digit, and evaluate on easier Sudokus with 24-36 givens. We also tried to train a version of our network that only had a loss at the last step. It was harder to train, performed worse and didn't learn a convergent algorithm.

## 3.2   BABI

BaBi is a text based QA dataset from Facebook (Weston et al., 2015) designed as a set of toy prerequisite tasks for reasoning, and is widely used in the deep learning literature. It consists of 20 tasks including deduction, induction, spatial and temporal reasoning, etc. Each question, e.g. "where is john?" is preceded by a number of facts in the form of short sentences, e.g. "john went to the kitchen.". A task is considered solved if a model achieves greater than 95% accuracy. The most difficult tasks require three steps of relational reasoning. As such the relational reasoning required is limited.

The relational reasoning module needs to reason about the facts, in context of the questions so we consider each sentence a node in a fully connected graph. The sentences are encoded using a LSTM with 32 hidden units. The question is also encoded using a LSTM with 32 hidden units. We

| Method | Givens | Accuracy |
|---|---|---|
| *Recurrent Relational Network* (this work) | 17 | **96.6%** |
| Loopy BP, modified (Khan et al., 2014) | 17 | 92.5% |
| Loopy BP, random (Bauke, 2008) | 17 | 61.7% |
| Loopy BP, parallel (Bauke, 2008) | 17 | 53.2% |
| Deeply Learned Messages (Lin et al., 2015) | 17 | 0% |
| Relational Network, node (Santoro et al., 2017) | 17 | 0% |
| Relational Network, graph (Santoro et al., 2017) | 17 | 0% |
| Deep Convolutional Network (Park, 2016) | 24-36 | 70% |

Table 1: Comparison of methods for solving Sudoku puzzles. Only methods that are differentiable are included in the comparison.

concatenate the last hidden state of the sentence LSTM with the last hidden state of the question LSTM and pass that through a MLP. The output is considered the node features $x_i$. We set all edge features $e_{ij}$ to the question encoding following (Santoro et al., 2017). We only consider the preceding 20 sentences to a question. Our message function $f$ is identical to the Sudoku message function, i.e. a MLP which feeds into a LSTM. We run our network for five steps. To get a graph level output, we use a MLP over the sum of the node hidden states, with 3 layers, the final being a linear layer that maps to the output dimensionality logits. Unless otherwise specified we use 128 hidden units for all layers and all MLPs are 3 ReLU layers followed by a linear layer. We train on all the 10.000 training samples, using Adam with a batch size of 640, a learning rate of 2e-4 and L2 regularization with a rate of 1e-4.

Our trained network solves 19 of 20 tasks, which is competitive with state-of-the-art. Most tasks are quickly and perfectly learned. The only task that the network cannot complete is number 16, the induction task. See table 2 for the tasks where the model achieved less than 100% accuracy.

| Task | 2 | 3 | 5 | 14 | 16 | 18 | 19 |
|---|---|---|---|---|---|---|---|
| Accuracy | 99.7% | 96.5% | 99.6% | 99.9% | 45.1% | 99.7% | 99.9% |

Table 2: BaBi results. The tasks that are not shown are all 100% accurate.

On the BaBi task the relational network solves 18/20 tasks, notably failing on the 2 and 3 supporting fact tasks (Santoro et al., 2017). Training the relational network on BaBi takes millions of updates, and a couple of days on 10+ K80 GPUs (David Raposo, 2017, personal communication). In comparison our network naturally perform multi-step relational reasoning, and requires around half a million updates which takes approximately 12 hours on 4 Titan X GPUs. We hypothesize that the relational network takes longer to train because it cannot naturally perform multi-step relational reasoning.

End-to-end memory networks (Sukhbaatar et al., 2015) solves 14/20 tasks by using multiple recurrent hops of attention over the encoded sentences. The Sparse Differentiable Neural Computer (SDNC) is a differentiable computer modeled on the Turing Machine (Rae et al., 2016). It has a large external memory bank it updates by using sparse reads and writes. It solves 19/20 tasks which is state-of-the-art. It also fails at the induction task. EntNet reports 20/20 tasks solved, but does so training on each task independently. Trained jointly on all tasks EntNet solves 16/20 tasks (Henaff et al., 2016).

## 4    DISCUSSION

We have proposed a general relational reasoning model for solving tasks requiring an order of magnitude more complex relational reasoning than the current state-of-the art. It can be added to any deep learning model to provide a powerful relational reasoning capacity. We get state-of-the-art

results on Sudokus solving 96.6% of the hardest Sudokus with 17 givens. We also get results competitive with state-of-the-art results on the BaBi dataset solving 19/20 tasks.

Many difficult problems require complex relational reasoning and we see several exciting applications where our model might improve on state-of-the-art. Silver et al. (2017) mastered the game of Go with a deep residual network (He et al., 2016) with 79 convolutional layers in total that evaluate game position values and proposes moves combined with a monte-carlo tree search algorithm. It would be interesting to replace the deep residual network with our proposed model, and see if the capacity for complex relational reasoning could improve on AlphaGo. In a similar manner it might be possible to use it in a deep reinforcement learning setup and improve on the difficult Atari games that require long term planning and reasoning, e.g. Montezuma's revenge or Frostbite (Mnih et al., 2013). Finally we hypothesize it could improve on deep image captioning models (Karpathy & Fei-Fei, 2015) since reasoning about the people and objects involved in an image is essential to describing it.

Loopy belief propagation is widely used for performing approximate inference in graphical models with loops (Murphy et al., 1999). For the Sudoku problem our network learned an inference algorithm that outperforms loopy belief propagation, and it would be interesting to see if we could likewise improve on other problems that rely on loopy belief propagation. One prominent example of loopy belief propagation is in error correcting codes which everything from mobile phones to satellites rely on for robust communication (Shannon, 1948; MacKay & Neal, 1996).

## 5 RELATED WORK

Relational networks (Santoro et al., 2017) and interaction networks (Battaglia et al., 2016) are the most directly comparable to ours. They compare to using a single step of equation 3. Since it only does one step it cannot naturally do complex multi-step relational reasoning. In order to solve the tasks that require more than a single step it must compress all the relevant relations into a fixed size vector, then perform the remaining relational reasoning in the last forward layers. Both relational networks, interaction networks and our proposed model can be seen as an instance of Graph Neural Networks (Scarselli et al., 2009), (Gilmer et al., 2017). Our main contribution is showing how these can be used for complex relational reasoning.

Our model can be seen as a completely learned message passing algorithm. Belief propagation is a hand-crafted message passing algorithm for performing exact inference in directed acyclic graphical models. If the graph has cycles, one can use a variant, loopy belief propagation, but it is not guaranteed to be exact, unbiased or even converge. Empirically it works well though and it is widely used (Murphy et al., 1999). Several works have proposed replacing parts of belief propagation with learned modules (Heess et al., 2013; Lin et al., 2015). Our work differs by not being rooted in loopy BP, and instead learning all parts of a general message passing algorithm. Ross et al. (2011) proposes Inference Machines which ditch the belief propagation algorithm altogether and instead train a series of regressors to output the correct marginals by passing messages on a graph. Wei et al. (2016) applies this idea to pose estimation using a series of convolutional layers and Deng et al. (2016) introduces a recurrent node update for the same domain.

There is rich litterature on combining symbolic reasoning and logic with subsymbolic distributed representations which goes all the way back to the birth of the idea of parallel distributed processing McCulloch & Pitts (1943). See (Raedt et al., 2016; Besold et al., 2017) for two recent surveys. Here we describe only a few recent methods. Serafini & Garcez (2016) introduces the Logic Tensor Network (LTN) which describes a first order logic in which symbols are grounded as vector embeddings, and predicates and functions are grounded as tensor networks. The embeddings and tensor networks are then optimized jointly to maximize a fuzzy satisfiability measure over a set of known facts and fuzzy constraints. In Donadello et al. (2017) the LTN is used to improve on a Semantic Image Interpretation task by incorporating fuzzy prior constraints, e.g. cats usually have tails. Šourek et al. (2015) introduces the Lifted Relational Network which combines relational logic with neural networks by creating neural networks from lifted rules and training examples, such that the connections between neurons created from the same lifted rules shares weights. Our approach differs fundamentally in that we do not aim to bridge symbolic and subsymbolic methods. Instead we stay completely in the subsymbolic realm. We do not introduce or consider any explicit logic, aim to discover (fuzzy) logic rules, or attempt to include prior knowledge in the form of logical

constraints. The relational reasoning algorithm learned by our network is a black box to the extent that any neural network is a black box.

Amos & Kolter (2017) Introduces OptNet, a neural network layer that solve quadratic programs using an efficient differentiable solver. OptNet is trained to solve 4x4 Sudokus amongst other problems and beats the deep convolutional network baseline as described in Park (2016). Unfortunately we cannot compare to OptNet directly as it has computational issues scaling to 9x9 Sudokus due to an implementation error (Brandon Amos, 2018, personal communication).

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

## 6 APPENDIX

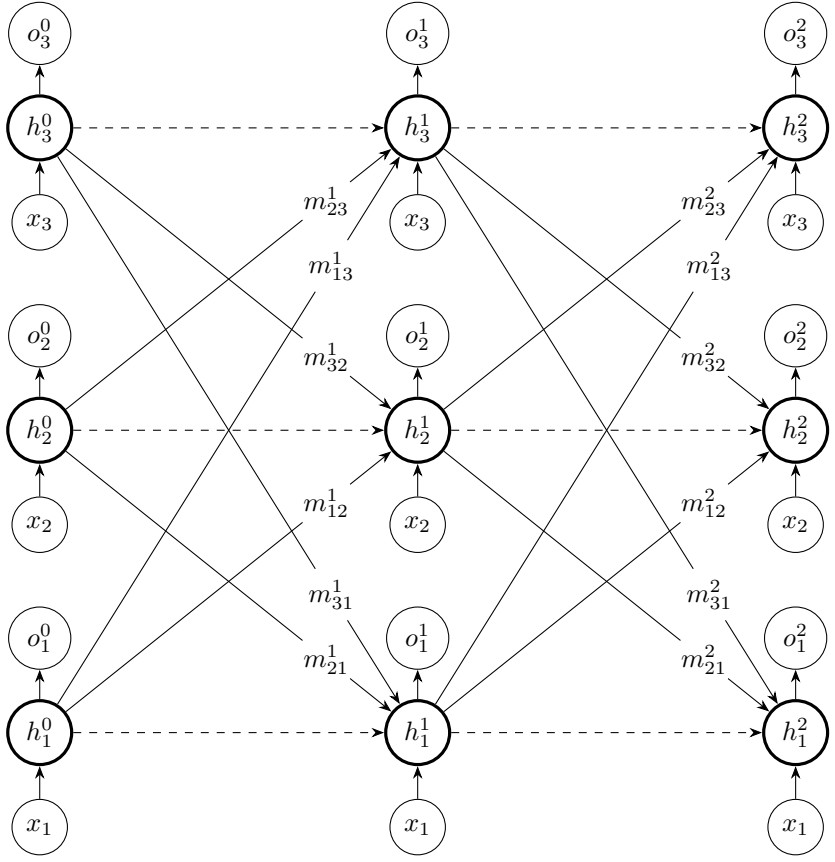

Recurrent relational network on a fully connected graph with 3 nodes. The same graph as in figure 1 unrolled over 2 steps. The node hidden states $h_i^t$ are highlighted. Subscripts denote node indices and superscripts denote steps $t$. The dashed lines indicate the recurrent connections.

## 7 NEURAL RELATIONAL REASONING INTERFACE

Given a graph $G$ specified by a set of $n$ nodes described by $d$-dimensional real valued vectors, $N \in \mathbb{R}^{n \times d}$ and $m$ directed edges $E \in \{1, 2, ..., n\}^{m \times 2}$, any function $f \colon \{N, E\} \to \mathbb{R}$ that is almost-everywhere differentiable respects the interface.

Given that the edges have attributes described by $p$-dimensional real valued vectors $A \in \mathbb{R}^{m \times p}$ any function $f \colon \{N, E, A\} \to \mathbb{R}$ that is almost-everywhere differentiable respects the interface.

Note that the functions map to a single real value. This might seem counter-intuitive. For the Sudoko problem for instance, the function must output a probability distribution over the digits for each of the 81 cells. But for the interface what matters is that the function output a single real valued loss, and be differentiable, such that it can be trained end-to-end with a perceptual front-end using stochastic gradient descent.

An example Sudoku. The full Sudoku from which the column in figure 2 is taken. Each of the 81 cells contain each digit 1-9, which is useful if the reader wishes to try to solve the Sudoku as they can be crossed out or highlighted, etc. The digit font size corresponds to the probability our model assigns to each digit at step 0, i.e. before any steps are taken. Subsequent pages contains the Sudoku as it evolves with more steps of our model.

The Sudoku grid below. Cells showing a single large digit indicate the model's high-probability assignment at step 0; cells showing the full set "1 2 3 / 4 5 6 / 7 8 9" indicate all candidate digits remain possible.

| | | | | | | | | |
|---|---|---|---|---|---|---|---|---|
| **4** | 1-9 | 1-9 | 1-9 | 1-9 | 1-9 | **6** | **9** | 1-9 |
| 1-9 | 1-9 | **3** | **2** | 1-9 | **4** | 1-9 | 1-9 | 1-9 |
| 1-9 | 1-9 | 1-9 | 1-9 | 1-9 | 1-9 | 1-9 | **1** | 1-9 |
| 1-9 | **3** | **9** | 1-9 | 1-9 | 1-9 | 1-9 | 1-9 | **2** |
| 1-9 | 1-9 | 1-9 | **8** | 1-9 | 1-9 | **5** | 1-9 | 1-9 |
| 1-9 | **1** | 1-9 | 1-9 | 1-9 | 1-9 | 1-9 | 1-9 | 1-9 |
| 1-9 | 1-9 | 1-9 | **9** | **1** | 1-9 | 1-9 | 1-9 | 1-9 |
| **8** | 1-9 | 1-9 | 1-9 | 1-9 | 1-9 | 1-9 | 1-9 | 1-9 |
| 1-9 | 1-9 | 1-9 | 1-9 | **6** | 1-9 | 1-9 | 1-9 | 1-9 |

Step 1

Step 4

| | | | | | | | | |
|---|---|---|---|---|---|---|---|---|
| 4 | 2 | 1 2 | 1 | 1 / 7 8 | 5 / 7 8 | 6 | 9 | 3 |
| 1 | 6 / 9 | 6 / 9 | 3 | 2 | 1 / 4 | 7 8 | 5 / 7 8 | 5 / 7 8 |
| 5 6 / 7 | 5 6 / 7 8 | 5 6 / 7 8 | 6 / 9 | 3 / 5 / 7 8 | 6 | 2 | 1 | 4 / 3 |
| 5 6 / 7 | 3 / 9 | | 1 / 4 6 / 7 | 1 / 4 5 / 7 | 5 6 / 7 | 1 / 4 / 7 8 | 4 6 / 7 8 | 2 |
| 2 / 6 / 7 | 2 / 4 6 / 7 | 2 / 4 6 / 7 | 8 | 1 2 3 / 4 / 7 | 2 3 / 6 / 9 | 5 | 3 / 4 6 / 7 | 1 / 6 / 9 |
| 2 / 5 6 / 7 | 1 | 8 | 3 / 4 5 6 / 7 9 | 2 3 / 4 5 / 7 | 2 3 / 5 6 / 7 9 | 3 / 4 / 7 8 9 | 3 / 4 6 / 7 8 | 3 / 4 6 / 7 8 9 |
| 2 3 / 5 6 / 7 | 2 / 4 5 6 / 7 | 2 / 4 5 6 / 7 | 3 / 4 5 / 7 9 | 1 | 2 3 / 4 5 6 / 7 8 | 2 3 / 4 5 6 / 7 8 | 4 5 6 / 7 8 |
| 8 | 2 / 4 5 6 / 7 9 | 1 2 / 4 5 6 / 7 | 3 / 4 5 / 7 | 2 3 / 4 5 / 7 | 2 3 / 5 / 7 | 1 / 4 / 9 | 2 3 / 4 5 6 / 7 | 1 / 6 / 9 |
| 1 / 3 / 9 | 2 / 4 5 / 7 9 | 1 2 / 4 5 / 7 | 3 / 4 5 / 7 | 6 | 2 / 8 | 1 2 3 / 4 / 7 8 9 | 2 3 / 4 5 / 7 8 | 1 3 / 4 5 / 7 8 9 |

Step 8

Step 12

Step 16

| | | | | | | | | |
|---|---|---|---|---|---|---|---|---|
| 4 | 2 | 1 | 7 | 8 | 5 | 6 | 9 | 3 |
| 9 | 6 | 3 | 2 | 1 | 4 | 7 | 8 | 5 |
| 7 | 8 | 5 | 9 | 6 | 3 | 2 | 1 | 4 |
| 5 | 3 | 9 | 1 | 4 | 7 | 8 | 6 | 2 |
| 6 | 4 7 | 4 7 | 8 | 2 | 9 | 5 | 3 | 1 |
| 2 | 1 | 8 | 6 | 5 | 9 | 3 | 4 | 7 |
| 5 7 | 3 | 2 5 6 7 | 5 | 9 | 1 | 4 | 2 | 6 8 |
| 8 | 4 5 9 | 4 6 9 | 3 | 7 | | 2 | 1 5 | 6 9 |
| 1 | 4 5 7 9 | 2 | 4 5 | | 6 8 | 3 | 2 | 7 9 |

Step 20

| 4 | 2 | 1 | 7 | 8 | 5 | 6 | 9 | 3 |
|---|---|---|---|---|---|---|---|---|
| 9 | 6 | 3 | 2 | 1 | 4 | 7 | 8 | 5 |
| 7 | 8 | 5 | 9 | 3 | 6 | 2 | 1 | 4 |
| 5 | 3 | 9 | 1 | 4 | 7 | 8 | 6 | 2 |
| 6 | 4 | 7 | 8 | 2 | 9 | 5 | 3 | 1 |
| 2 | 1 | 8 | 6 | 5 | 3 | 9 | 4 | 7 |
| 3 | 7 | 6 | 5 | 9 | 1 | 4 | 2 | 8 |
| 8 | 4 | 9 | 3 | 7 | 2 | 1 | 5 | 6 |
| 1 | 5 | 2 | 4 | 6 | 8 | 3 | 7 | 9 |

