# OpenReview forum: "Recurrent Relational Networks for complex relational reasoning"
_ICLR.cc/2018/Conference — Reject_

### Official Review · AnonReviewer2 · 2017-11-25
**Very nice direction but no complex relational reasoning demonstrated and missing related work**

**Rating:** 3
**Confidence:** 5

**Review:**

The paper introduced recurrent relational network (RRNs), an enhanced version of the
existing relational network, that can be added to any neural networks to add
relational reasoning capacity. RRNs are illustrated on sudoku puzzles and textual QA.

Overall the paper is well written and structured. It also addresses an important research question: combining relational reasoning and neural networks is currently receiving a lot of attention, in particular when generally considering the question of bridging sub-symbolic and symbolic methods. Unfortunately, it is current form, the paper has two major downsides. First of all,  the sudoku example does not illustrate “complex relational reasoning” as claimed in the title. The problem is encoded at a positional level where
messages encoded as MLPs and LSTMs implement the constraints for sudoko. Indeed,
this allows to realise end-to-end learning but does not illustrate complex reasoning.
This is also reflected in the considered QA task, which is essentially coded as a positional problem. Consequently, the claim of the conclusions, namely that “we have
proposed a general relational reasoning model” is not validated, unfortunately. Such
a module that can be connected to any existing neural network would be great. However,
for that one should show capabilities of relational logic. Some standard (noisy)
reasoning capabilities such as modus ponens. This also leads me to the second downside.
Unfortunately, the paper falls short on discussion related work. First of all,
there is the large field of statistical relational learning, see

Luc De Raedt, Kristian Kersting, Sriraam Natarajan, David Poole:
Statistical Relational Artificial Intelligence: Logic, Probability, and Computation. Synthesis Lectures on Artificial Intelligence and Machine Learning, Morgan & Claypool Publishers 2016

for a recent overview. As it has the very same goals, while not using a neural architecture for implementation, it is very much related and has to be discussed. That
one can also use a neural implementation can be seen in

Ivan Donadello, Luciano Serafini, Artur S. d'Avila Garcez:
Logic Tensor Networks for Semantic Image Interpretation. IJCAI 2017: 1596-1602

Matko Bosnjak, Tim Rocktäschel, Jason Naradowsky, Sebastian Riedel:
Programming with a Differentiable Forth Interpreter. ICML 2017: 547-556

Luciano Serafini, Artur S. d'Avila Garcez:
Learning and Reasoning with Logic Tensor Networks. AI*IA 2016: 334-348

Gustav Sourek, Vojtech Aschenbrenner, Filip Zelezný, Ondrej Kuzelka:
Lifted Relational Neural Networks. CoCo@NIPS 2015

Tim Rocktäschel, Sebastian Riedel:
End-to-end Differentiable Proving. CoRR abs/1705.11040 (2017)

William W. Cohen, Fan Yang, Kathryn Mazaitis:
TensorLog: Deep Learning Meets Probabilistic DBs. CoRR abs/1707.05390 (2017)

to list just some approaches. There are also (deep) probabilistic programming
approaches such as Edward that should be mentioned as CPS like problems (Sudoku) can
definitely be implement there. Moreover, there is a number of papers that discuss
embeddings of relational data and rules such as

William Yang Wang, William W. Cohen:
Learning First-Order Logic Embeddings via Matrix Factorization. IJCAI 2016: 2132-2138

Thomas Demeester, Tim Rocktäschel, Sebastian Riedel:
Lifted Rule Injection for Relation Embeddings. EMNLP 2016: 1389-1399

and even neural-symbolic approaches with a long publication history. Unfortunately,
non of these approaches has been cited, giving the wrong impression that this is
the first paper that tackles the long lasting question of merging sub-symbolic and symbolic reasoning. BTW, there have been also other deep networks for optimisation, see e.g.

Brandon Amos, J. Zico Kolter:
OptNet: Differentiable Optimization as a Layer in Neural Networks.
ICML 2017: 136-145

that have also considered Sudoku. To summarise, I like very much the direction of the paper but it seems to be too early to be published.

---

> ### Author Response · Authors · 2017-12-07
> **Disagree what constitute reasoning**
>
> Thank you for the review.
>
> > Overall the paper is well written and structured. It also addresses an important research question: combining relational reasoning and neural networks is currently receiving a lot of attention, in particular when generally considering the question of bridging sub-symbolic and symbolic methods.
>
> Answer: Thank you.
>
> > Unfortunately, it is current form, the paper has two major downsides.
>
> Answer: We get the objections put forward below. Our terminology clearly has made you expect a paper on statistical relational learning. We believe that we are solving problems that requires what we associate with relational reasoning although that it does not involve explicit “relational logical” in the first order logic sense. So in short we think it is acceptable to view reasoning in a broader sense as also done by Santoro et al. Detailed answers below.
>
> > First of all, the sudoku example does not illustrate “complex relational reasoning” as claimed in the title. The problem is encoded at a positional level where messages encoded as MLPs and LSTMs implement the constraints for Sudoko. Indeed, this allows to realise end-to-end learning but does not illustrate complex reasoning. This is also reflected in the considered QA task, which is essentially coded as a positional problem.
>
> Answer: Clearly, there are ample opportunity for misunderstandings given how ambiguous notions such as “complex”, “relational” and “reasoning” are. For your definitions of these concepts you are right that our network does not perform it. Obviously with our definitions, we think it does, otherwise we wouldn’t have made those experiments or claims. So our definitions are almost certainly different.
>
> Our use of the term “relational reasoning” follows Santoro et al. and is to be honest quite vague. By “relational reasoning” we mean to represent the world as, and perform inference over, a set of objects and their relations. We did not intend to claim that we are performing “relational logic” in the strict first-order logic sense, e.g. with variables and quantifiers. Is this the source of the disagreement? If so, we’re happy to amend our paper to make this more clear. If not would you be kind enough to clarify your definitions of those concepts such that it is immediately obvious that our network does not perform “relational reasoning” under those definitions? Also, specifically could you clarify what you mean by “positional encoding/problem” and how that nullifies “reasoning”?
>
> > Consequently, the claim of the conclusions, namely that “we have proposed a general relational reasoning model” is not validated, unfortunately. Such a module that can be connected to any existing neural network would be great. However, for that one should show capabilities of relational logic. Some standard (noisy) reasoning capabilities such as modus ponens.
>
> Answer: Sudoku can be formulated as a logical problem using e.g. propositional logic or first-order logic, and solved using e.g. SAT a solver. It's clearly a logical problem that requires reasoning to solve (efficiently). Are you really arguing that solving Sudoku does not require reasoning? Also, w.r.t modus ponens, the first step of our RRN eliminates digits which demonstrates (fuzzy) modus ponens in which “x implies (not y), x, thus (not y)”. It’s not exact logic, and you can’t extract the logical clauses and inspect them, as in many SRL systems, but that does not mean it’s not reasoning or useful. Just like human reasoning is inexact and impossible to introspect, but still reasoning and very useful.
>
> > This also leads me to the second downside. Unfortunately, the paper falls short on discussion related work. First of all, there is the large field of statistical relational learning, see
>
> Answer: Thank you for these references. These interesting papers are trying to solve different problems than those we consider. We will discuss these and clarify this in the updated paper.

---

### Official Review · AnonReviewer3 · 2017-11-27
**Interesting, clearly presented new structured prediction algorithm. Paper marginally below acceptance threshold.**

**Rating:** 5
**Confidence:** 3

**Review:**

This paper introduces recurrent relational networks: a deep neural network for structured prediction (or relational reasoning). The authors use it to achieve state-of-the-art performance on Soduku puzzles and the BaBi task (a text based QA dataset designed as a set of to toy prerequisite tasks for reasoning).

Overall I think that by itself the algorithm suggested in the paper is not enough to be presented in ICLR, and on the other hand the authors didn't show it has a big impact (could do so by adding more tasks - as they suggest in the discussion). This is why I think the paper is marginally below the acceptance threshold but could be convinced otherwise.

C an the authors give experimental evidences for their claim: "As such, the network could use a small part of the hidden state for retaining a current best guess, which might remain constant over several steps, and other parts of the hidden state for running a non-greedy..." -

Pros
- The idea of the paper is clearly presented, the algorithm is easy to follow.
- The motivation to do better relational reasoning is clear and the network suggested in this paper succeeds to achieve it in the challenging tasks.

Cons
- The recurrent relational networks is basically a complex learned message passing algorithm. As the authors themselves state there are several works from recent years which also tackle this (one missing reference is Deeply Learning the Messages in Message Passing Inference of Lin et al from NIPS 2016). It would been interesting to compare results to these algorithms.
- For the Sudoku the proposed architecture of the network seems a bit to complex, for example why do a 16 embedding is needed for representing a digit between 0-9? Some other choices (batch size of 252) seem very specific.

---

> ### Author Response · Authors · 2017-12-07
> **Answer**
>
> Thank you for the review. And thank you for the kind words regarding motivation and clarity.
>
> >Overall I think that by itself the algorithm suggested in the paper is not enough to be presented in ICLR, and on the other hand the authors didn't show it has a big impact (could do so by adding more tasks - as they suggest in the discussion).
> This is why I think the paper is marginally below the acceptance threshold but could be convinced otherwise.
>
> Answer: The presented algorithm is a plug-n-play neural network module for solving problems requiring complex reasoning. We clearly show how it can solve a difficult reasoning problem, Sudoku, which comparable state-of-the-art methods cannot. We also show state-of-the-art results on a very different task, BaBi, showing its general applicability. We’ve released the code (but can’t link it yet, due to double blind). Simply put, with this algorithm the community can approach a swathe of difficult reasoning problems, which they couldn’t before. As such we think it merits publication. If you’re not convinced, what additional task would make you excited about this algorithm?
>
> >Can the authors give experimental evidences for their claim: "As such, the network could use a small part of the hidden state for retaining a current best guess, which might remain constant over several steps, and other parts of the hidden state for running a non-greedy..." -
>
> Answer: To clarify, we are not claiming that it does this, just that it has the capacity since the output and the hidden state is separated by an arbitrarily complex function. Since the function is arbitrarily complex it can learn to conditionally ignore parts of the hidden state, similar to how a LSTM can learn to selectively update its memory. We don’t have any experimental evidence whether it actually does this.
>
> > The recurrent relational networks is basically a complex learned message passing algorithm. As the authors themselves state there are several works from recent years which also tackle this (one missing reference is Deeply Learning the Messages in Message Passing Inference of Lin et al from NIPS 2016). It would been interesting to compare results to these algorithms.
>
> Answer: Yes, that’s a fair point. Comparing to those methods would be interesting. We know the Lin et al.’s paper but somehow forgot to cite it. It will be added in the update. One important difference is that those works retain parts of loopy belief propagation and learn others whereas ours is completely learned.
>
> > For the Sudoku the proposed architecture of the network seems a bit too complex, for example why do a 16 embedding is needed for representing a digit between 0-9? Some other choices (batch size of 252) seem very specific.
>
> Answer: We’re reporting all the gory details out of a (misplaced?) sense of scientific rigor, not because they are important hyper parameters. The 16 dimensional embedding was simply the first thing we tried. It’s not the result of extensive hyper-parameter tuning. We could have used a one-hot encoding, but then the next matrix multiply would effectively be the embedding. We think it’s a bit cleaner to have the embedding separately. Also using an embedding the x vector more closely resembles the expected input to the RRN from a perceptual front-end, e.g. a dense vector. The 252 batch size was simply so that the batch size was divisible by 6, because we trained on 6 GPUs.

---

### Official Review · AnonReviewer1 · 2017-11-28
**The proposed method should be better explained**

**Rating:** 5
**Confidence:** 3

**Review:**

This paper describes a method called relational network to add relational reasoning capacity to deep neural networks. The previous approach can only perform a single step of relational reasoning, and was evaluated on problems that require at most three steps. The current method address the scalability issue and can solve tasks with orders of magnitude more steps of reasoning. The proposed methods are evaluated on two problems, Sudoku and Babi, and achieved state-of-the-art results.

The proposed method should be better explained. What’s the precise definition of interface? It’s claimed that other constraint propagation-based methods can solve Sudoku problems easily, but don’t respect the interface. It is hard to appreciate without a precise definition of interface. The proposed recurrent relational networks are only defined informally. A definition of the model as well as related algorithms should be defined more formally.

---

> ### Author Response · Authors · 2017-12-07
> **Explanation**
>
> Thank you for the review.
>
> > The proposed method should be better explained. What’s the precise definition of interface? It’s claimed that other constraint propagation-based methods can solve Sudoku problems easily, but don’t respect the interface.
>
> Answer: A function respecting the interface must accept a graph (set of nodes and set of edges) where the nodes are described with real valued vectors and most importantly output a solution which is *differentiable* w.r.t. the parameters of the function.
> Traditional Sudoku solvers, e.g. constraint propagation and search are not differentiable, since they use non-differentiable operations e.g. hard memory lookups, writes, if statements, etc.
>
> We need the function to respect this interface so it can be used with other neural network modules, and trained end-to-end. For an example see “A simple neural network module for relational reasoning” in which a Relation Network is added to a Convolutional Neural Network and trained end-to-end to reason about objects in images. Similarly with our BaBi example we combine a LSTM that reads each sentence with a Recurrent Relational Network to reason about the sentences.
>
> >It is hard to appreciate without a precise definition of interface. The proposed recurrent relational networks are only defined informally. A definition of the model as well as related algorithms should be defined more formally.
>
> Answer: The first draft of the paper actually had a very formal introduction of the interface and algorithm, but we re-worked it to the current informal, example-driven style since we thought it was easier to understand and follow. We’ll add the rigorous definition to the appendix. We’ll let you know once we’ve updated the paper.

---

### Decision · Program_Chairs · 2018-01-29
**ICLR 2018 Conference Acceptance Decision**

**Decision:**

Reject

**Comment:**

The proposed relational reasoning algorithm is basically a fairly standard graph neural network, with a few modifications (e.g., the prediction loss at each layer - also not a new idea per se).

 The claim that previously reasoning has not been considered in previous applications of graph neural networks (see discussion) is questionable.  It is not even clear what is meant here by 'reasoning' as many applications of graph neural networks may be regarded as performing some kind of inference on graphs (e.g., matrix completion tasks by Berg, Kipf and Welling; statistical relational learning by  Schlichtkrull et al).

So the contribution seems a bit over-stated.  Rather than introduces a new model, the work basically proposes an application of largely known model to two (not-so-hard) tasks which have not been studied in the context of GNNs. The claim that the approach is a general framework for dealing with complex reasoning problems is not well supported as both problems are (arguably) not complex reasoning problems (see R2).

There is a general consensus between reviewers that the paper, in its current form, does not quite meet acceptance criteria.

Pros:
-- an interesting direction
-- clarity
Cons:
-- the claim of generality is not well supported
-- the approach is not so novel
-- the approach should be better grounded in previous work